# Usefulness of Procalcitonin in the Diagnosis of Bacterial Infection in Immunocompetent Children

**DOI:** 10.3390/children9081263

**Published:** 2022-08-21

**Authors:** Hae Na Park, Su Yeong Kim, Na Mi Lee, Dae Yong Yi, Sin Weon Yun, Soo Ahn Chae, In Seok Lim, Yong Kwan Lim, Ji Young Park

**Affiliations:** 1Department of Pediatrics, Chung-Ang University Hospital, Seoul 06973, Korea; 2Department of Laboratory Medicine, Chung-Ang University Hospital, Seoul 06973, Korea

**Keywords:** bacterial infections, child, procalcitonin

## Abstract

Bacterial infections (BIs) need to be differentiated from non-BIs (NBIs) to enable prompt administration of antibiotics. Therefore, inflammatory biomarkers are needed as they can accurately identify BIs. This study evaluated the usefulness of procalcitonin (PCT) in the diagnosis of BI in immunocompetent children. We retrospectively reviewed the medical records of patients <18 years who underwent PCT measurements between July 2012 and June 2019. In total, 474 patients were enrolled and divided into the BI (n = 205) and NBI groups (n = 269). The BI group was subcategorized into the invasive BI (IBI; n = 94), mucosal BI (MBI; n = 31), toxigenic BI (TBI; n = 23), and localized BI (LBI; n = 57) subgroups. The NBI group was further subcategorized into the viral infection (VI; n = 118) and inflammatory disease groups (ID; n = 151). PCT was compared with the levels of C-reactive protein (CRP), white blood cell (WBC), and erythrocyte sedimentation rate (ESR). Between the BI and NBI groups, PCT (4.2 ± 16.9 vs. 1.1 ± 2.5 ng/mL; *p* = 0.008) and ESR (39.1 ± 32.4 vs. 54.8 ± 28.2 mm/h; *p* < 0.001) were significantly different. Between the IBI and other groups, WBC (14,797 ± 7148 vs. 12,622 ± 5770 × 10^6^/L; *p* = 0.007), ESR (35.3 ± 30.3 vs. 51.5 ± 30.3 mm/h; *p* < 0.001), and PCT (8.1 ± 23.8 vs. 1.0 ± 3.4 ng/mL; *p* = 0.005) were significantly different. However, none of the biomarkers were useful in differentiating BI from NBI. While WBC (area under curve (AUC) = 0.615, *p* = 0.003) and PCT (AUC = 0.640, *p* < 0.001) were useful, they fared poorly in differentiating IBI from other groups. Thus, additional studies are needed to identify more accurate biomarkers capable of differentiating BIs, especially IBIs.

## 1. Introduction

In pediatric patients, viral infections (VIs) are the main causes of infectious diseases. However, because clinicians find it troublesome to differentiate bacterial infections (BIs) from VIs with initial clinical presentations for many conditions, empirical antibiotic therapy is administered. To avoid unnecessary antibiotic use, diagnostic inflammatory markers that can suggest the occurrences of BIs quickly and accurately have been a requirement [1]. Administration of antibiotics 1 h after visiting the emergency department could lower mortality [2]. Thus, in addition to the clinician’s alertness to the clinical presentations, an adequate inflammatory marker that suggests possible BI is needed. Inflammatory markers, including white blood cell (WBC) count, erythrocyte sedimentation rate (ESR), or C-reactive protein (CRP) have been extensively used.

Procalcitonin (PCT), a pro-peptide of calcitonin, is normally secreted from the C-cells of the thyroid gland [3]. It is usually secreted at a low serum concentration (<0.1 ng/mL) in normal adults and children after the neonatal period. In severe BI cases, PCT could increase from 0.5 ng/mL to 200 ng/mL. PCT rarely exceeds 1 ng/mL in cases of VI [4]. The half-life time of PCT is approximately 24 h [5]. Although the mechanism behind the elevation of PCT levels during BI is yet to be identified, the most extensively accepted hypothesis states that PCT is secreted from hematopoietic cells, parenchyma of the lung, or the liver by proinflammatory cytokines, such as interleukin (IL)-1, IL-8, and tissue necrosis factor α post BI [6]. In South Korea, studies regarding PCT were initiated in the late 1990s [7,8]. Over the past decades, PCT has gradually been introduced in the clinical praxis as an inflammatory marker to differentiate various types of BI [9], and has been used along with other markers. Although early studies have shown the potential of PCT in differentiating various types of BI, recent studies have reported that PCT has no significant benefit when compared with CRP [10,11]. However, PCT had been recently studied again as an inflammatory marker to rule out invasive BI (IBI) [12,13].

Therefore, this study aims to evaluate the usefulness of PCT in the diagnosis of BI in children, especially IBI, by comparing it with other inflammatory markers.

## 2. Materials and Methods

### 2.1. Participants and Definition of Patient Groups

This retrospective analysis was conducted among immunocompetent patients under the age of 18 years who underwent a PCT test in the outpatient department or emergency department or as inpatients on the day of their admission to the Chung-Ang University Hospital between July 2012 and June 2019. The age group was classified into infants (<1 year, toddlers (1–2 years), preschoolers (3–5 years), middle childhood (6–11 years), and teenagers (12–17 years). We classified patients into BI and non-BI (NBI) groups, and further categorized the BI group into four subgroups: IBI, mucosal BI (MBI), toxigenic BI (TBI), and localized BI (LBI) according to the etiologies with final diagnosis.

The IBI group comprised patients in whom bacteria were identified by culture studies in blood, urine collected through aseptic catheterization or cystocentesis, and cerebrospinal fluid (CSF) samples [14]. The MBI group included patients with acute otitis media, acute sinusitis, and group A streptococcal pharyngotonsillitis. In the TBI group, patients with scarlet fever, staphylococcal scaled skin syndrome (4S), toxic shock syndrome (TSS), rheumatic fever, and acute generalized exanthematous pustulosis were included. In the LBI group, patients with skin and soft tissue infection, deep neck infection, mastoiditis, intra-abdominal infection, and bacterial enteritis were included. The NBI group was further divided into VI and inflammatory disease (ID) subgroups. The VI group included patients in whom viruses were detected through respiratory specimens, CSF, or stool, based on the polymerase chain reaction (PCR) or rapid antigen test. Patients with combined bacterial and viral etiologies were excluded. The ID group included patients with autoimmune or non-infectious IDs. Patients in the ID group with positive bacterial or viral studies were also excluded.

This study was approved by the Institutional Review Board of Chung-Ang University Hospital (no. 2004-002-19308). The review board waived the need for informed consent.

### 2.2. Study Design

All patients with data pertaining to PCT, CRP, WBC count, and ESR were included. We considered additional laboratory tests depending on the patient’s symptoms and physical examination. If a patient had gastrointestinal symptoms, such as vomiting and diarrhea, we performed a stool virus antigen/PCR test or conducted a culture study. Nasopharyngeal swabs were collected from patients with respiratory symptoms, such as cough, sputum, or rhinorrhea, which were further processed for the detection of respiratory viruses (adenovirus, respiratory syncytial virus A/B, Influenza A/B virus, parainfluenza virus 1/2/3/4, rhinovirus A/B/C, metapneumovirus, enterovirus, coronavirus 229E/NL63/OC43, and bocavirus 1/2/3/4) via PCT test. If patients exhibited signs of meningeal irritation or if infants under 90 days had fever, they were subjected to CSF analysis with culture study and PCR tests for enterovirus/herpes virus. The final diagnosis was made based on the laboratory results. The presence of a single bacterial pathogen in stool specimens indicated bacterial enteritis and the presence of a single viral pathogen in stool specimens indicated viral enteritis. Patients with WBC count ≥5 (based on high-power microscopy) and single bacterial pathogen ≥105 colony/mL (based on cultures) were diagnosed with urinary tract infection (UTI) Cases with WBC count ≥5 (based on high-power microscopy) and single bacterial pathogens ≥10^5^ colony forming unit/mL (based on cultures) were defined as urinary tract infection (UTI). The presence of bacterial pathogens in CSF culture indicated bacterial meningitis, while the presence of viral pathogens in CSF specimens, detected via PCR test, indicated viral meningitis. Pathogenic single bacterial infection identified by blood culture studies from more than two blood samples was defined as bacteremia. Scarlet fever, 4S, TSS, and acute generalized exanthematous pustulosis were diagnosed by their characteristic features. Rheumatic fever was diagnosed using Jones criteria. Group A streptococcal pharyngotonsillitis was diagnosed only if confirmed by a positive rapid antigen detection test or culture study. In our institute, the reference cut-off value and the measuring range for PCT were 0.5 ng/mL and 0.05–200 ng/mL, respectively.

### 2.3. Statistical Analysis

Statistical software programs SPSS (version 26.0, IBM, Armonk, NY, USA) and MedCalc 12, were used for statistical analysis. Categorical variables were compared using Pearson’s chi-square test. Continuous variables were compared using Student’s *t*-test and one-way analysis of variance. Continuous variables are presented as means ± standard deviations. A receiver operating characteristic (ROC) curve analysis was performed to obtain appropriate cut-off values, sensitivity, and specificity of inflammatory markers. ROC curve comparisons were also performed for each model set by using Delong’s test for two correlated ROC curves. Confidence intervals (CI) at 95% were verified and *p* values < 0.05 were considered statistically significant. 

## 3. Results

### 3.1. Demographics and Diagnosis of Patients

During the 7-year study period, 24,691 immunocompetent children under 18 years of age visited our medical center. Among them, 3203 children underwent PCT tests in our emergency department (n = 350), outpatient clinic (n = 577), and hospitalized ward (n = 2276). Overall, 1567 cases were revealed to be follow-up cases and were excluded. Additionally, 193 neonatal intensive care unit cases were excluded. Moreover, 952 children who did not undergo PCR tests were excluded due to unclarified etiologies. Furthermore, 17 cases were excluded because they were combined BI/VI cases or were IDs with VIs. Finally, 474 patients were enrolled in this study. All demographic data are listed in Table 1. In total, 197 patients were females (41.6%) with a mean age of 3.7 ± 4.0 years. Among the 462 admitted patients, the mean duration of hospitalization was 6.7 ± 7.1 days.

Two-hundred and five patients (43.2%) were classified into the BI group and 269 (56.8%) into the NBI group. The BI group was further subclassified into the IBI (n = 94; 45.9%), LBI (n = 57; 27.8%), MBI (n = 31; 15.1%), and TBI groups (n = 23; 11.2%). The NBI group was subclassified into the ID (n = 151; 56.1%) and VI groups (n = 118; 43.9%). The IBI group included patients with UTI (n = 73; 77.7%), bacteremia (n = 12; 12.8%), bone and joint infections (n = 5; 5.3%), and meningitis (n = 4; 4.3%). The MBI group included patients with acute otitis media or sinusitis (n = 23; 74.2%) and group A streptococcus pharyngotonsillitis (n = 8; 25.8%). The TBI group consisted of patients with scarlet fever (n = 12; 52.2%), 4S (n = 7; 30.4%), and TSS (n = 2; 8.7%); one patient (4.4%) had rheumatic fever and another had acute generalized exanthematous pustulosis. The LBI group included patients with skin and soft tissue infections (n = 21; 36.2%), acute cervical lymphadenitis (n = 14; 24.1%), intra-abdominal infection (n = 13; 22.4%), bacterial enteritis (n = 6; 10.3%), deep-neck infections (n = 2; 3.5%), and mastoiditis (n = 2; 3.5%). In the ID group, Kawasaki disease was the most common (n = 137; 90.7%), followed by Kikuchi–Fujimoto disease (n = 8; 5.3%), juvenile rheumatoid arthritis (n = 5; 3.3%), and ulcerative colitis (n = 1; 0.7%).

The number of females in the BI and NBI groups was 83 (40.5%) and 114 (42.4%), respectively, with no significant differences between the groups (*p* = 0.679). Between the two groups, there was no significant difference in age (BI: 4.2 ± 5.0 years, NBI: 3.4 ± 3.0 years; *p* = 0.057); however, a significantly longer hospitalization was observed in the patients in the BI group (8.7 ± 10.1 days) than those in the NBI group (5.3 ± 2.7 days; *p* < 0.001). There was no statistical sex-based difference among the BI, VI, and ID groups (females in BI: 83 [40.5%], in VI: 49 [41.5%], in ID: 65 [43.0%]; *p* = 0.889); however, significant differences were noted with respect to age (BI: 4.2 ± 5.0, VI: 2.9 ± 2.6, ID: 3.8 ± 3.2 years; *p* = 0.027) and duration of hospitalization (BI: 8.7 ± 10.1, VI: 5.1 ± 3.1, ID: 5.5 ± 2.4 days; *p* < 0.001). Upon comparison of the three groups, age differences were observed between the BI and VI groups (*p* = 0.020). Additionally, differences in the duration of hospitalization were observed between the BI and VI groups and the BI and ID groups (all *p* < 0.001). (Table 1) When comparing the IBI group with the other groups, sex-based differences were not observed (*p* = 0.342). However, the mean age was lower (1.5 ± 3.3 vs. 4.3 ± 4.0 years; *p* < 0.001) and the mean duration of hospitalization was higher in the IBI group (10.1 ± 11.0 vs. 5.8 ± 5.3 days; *p* = 0.001). A comparison among the BI subgroups revealed no significant difference with respect to sex (*p* = 0.360). The mean age and hospitalization duration were significantly different among the BI subgroups (*p* < 0.001 and *p* = 0.033, respectively). Detailed comparisons among the BI subgroups revealed that the mean age of patients in the IBI group (1.5 ± 3.3 years) was significantly lower than that of patients in MBI (4.4 ± 4.5 years, *p* = 0.008), TBI (5.8 ± 3.9 years, *p* < 0.001), and LBI groups (7.8 ± 5.5 years, *p* < 0.001). Patient age in the MBI and LBI groups was also significantly different (*p* = 0.002). The mean duration of hospitalization was different between the IBI and MBI groups (10.1 ± 11.0 vs. 3.8 ± 1.7 days, *p* = 0.026). The mean duration of hospitalization was 6.6 ± 4.3 days in the TBI group and 7.8 ± 5.5 days in the LBI group.

### 3.2. Analysis of Inflammatory Markers in Studied Groups

The mean value was 13,053 ± 6121 *×* 10^6^/L for WBC, 61.9 ± 64.4 mg/L for CRP, 48.1 ± 31.0 mm/h for ESR, and 2.4 ± 11.4 ng/mL for PCT. The mean values of inflammatory biomarkers by groups and subgroups are listed in Table 2. Among the inflammatory markers, there was a significant difference in PCT (BI: 4.2 ± 16.9, NBI: 1.1 ± 2.5 ng/mL; *p* = 0.008) and ESR (BI: 39.1 ± 32.4, NBI: 54.8 ± 28.2 mm/h; *p* < 0.001) between the BI and NBI groups. When comparing the differences in inflammatory markers among the BI, VI, and ID groups, there were significant differences in WBC, CRP, ESR, and PCT (*p* = 0.017, <0.001, <0.001, and 0.011, respectively). WBC values were significantly different between the VI and ID groups (11,767 ± 6207 vs. 13,890 ± 5378 × 10^6^/L; *p* = 0.013). CRP values were also significantly different in each group (BI: 61.4 ± 69.8, VI: 40.7 ± 51.5, ID: 79.0 ± 61.1 mg/L; all *p* < 0.05). ESR values in the ID group (61.4 ± 28.0 mm/h) were significantly different from that in the VI (38.3 ± 21.4 mm/h; *p* < 0.001) and BI groups (39.1 ± 32.4 mm/h; *p* < 0.001). The PCT values in the BI group (4.2 ± 16.9 ng/mL) were different from that in the VI (1.0 ± 3.3 ng/mL; *p* = 0.040) and ID groups (1.1 ± 1.7 ng/mL; *p* = 0.026). When compared, the WBC count (14,797 ± 7148 vs. 12,622 ± 5770 × 10^6^/L; *p* = 0.007) and PCT (8.1 ± 23.8 vs. 1.0 ± 3.4 ng/mL; *p* = 0.005) were found to be higher in the IBI group than in the other groups. Additionally, ESR was lower (35.3 ± 30.3 vs. 51.5 ± 30.3 mm/h; *p* < 0.001) in the IBI group than in the other groups, while CRP in the IBI group was not significantly different from the other groups (69.3 ± 77.6 vs. 60.0 ± 60.6 mg/L; *p* = 0.207). When analyzing the inflammatory markers among the BI subgroups, there were significant differences in WBC count and PCT (*p* = 0.011 and *p* = 0.025, respectively); however, CRP and ESR did not yield any significant difference (*p* = 0.060 and 0.357, respectively). Additionally, WBC count (14,797 ± 7148 vs. 11,941 ± 6203 × 10^6^/L) and PCT comparisons (8.1 ± 23.8 vs. 0.6 ± 1.1 ng/mL) between the IBI and LBI groups yielded significant differences (all *p* = 0.040).

There were significant differences according to the age groups between the BI and NBI groups and among the BI subgroups (all *p* < 0.001). Among infants, ESR alone was different between the BI and NBI groups (31.4 ± 27.6 vs. 48.1 ± 24.6 mm/h, *p* = 0.008) and the IBI and other groups (31.6 ± 27.5 vs. 46.1 ± 25.7 mm/h, *p* = 0.019). Among toddlers, no biomarker showed significant differences between the BI and NBI groups or the IBI and other groups (all *p* > 0.05). Among preschoolers, ESR was different between the BI and NBI (38.3 ± 34.8 vs. 58.1 ± 29.5 mm/h, *p* = 0.013) groups, while WBC was different between the IBI and other groups (19,295 ± 4280 vs. 12,782 ± 5206 × 10^6^/L, *p* = 0.015). Among middle childhood and teenagers, there was no difference between the BI and NBI groups or the IBI and other groups (all *p* > 0.05).

### 3.3. Comparisons of Accuracy and Cut-Off Values of Inflammatory Markers

The ROC curves were constructed by differentiating the BI and NBI groups. The areas under the curve (AUCs) were 0.495 for WBC, 0.443 for CRP, 0.335 for ESR, and 0.498 for PCT. All these markers were insignificant in distinguishing between the BI and NBI groups. (Table 3 and Figure 1A) However, when comparing AUCs between the IBI and other groups, PCT (0.640, 95% CI: 0.562–0.719, *p* < 0.001), WBC (0.615, 95% CI: 0.532–0.698, *p* = 0.003), and CRP (0.508, 95% 0.434–0.581, *p* = 0.843) were determined to be useful markers; however, CRP outcomes were not statistically significant. The optimal cut-off values were 15,860 × 10^6^/L for WBC (sensitivity: 46.8%, specificity: 76.3%) and 1.8 ng/mL for PCT (sensitivity: 39.4%, specificity: 87.1%). However, PCT and WBC yielded poor accuracy (Table 3 and Figure 1B).

## 4. Discussion

In this retrospective study, we compared inflammatory biomarker levels in children with BI and NBI and reported that PCT and ESR were significantly different in these groups. WBC, ESR, and PCT yielded significant differences, which aided in the differentiation of IBI from other infections. However, ROC analysis revealed that none of the four inflammatory markers were useful in differentiating BI from NBI. However, WBC and PCT were useful in differentiating IBI from other infections but yielded poor accuracy based on ROC analysis. ESR was not shown to be useful in the ROC analysis. Additionally, WBC and PCT yielded lower sensitivity but higher specificity than CRP and ESR.

Given that PCT can be measured to differentiate patients with LBI or VI from those with severe BI [9], many studies have been published in support of this hypothesis. According to several studies, PCT reflects the severity of BI and is more elevated in cases of severe BI. Additionally, a study with 175 children revealed that patients with septic shock had higher PCT levels than those with no infection, VI, or LBI and that PCT values were further increased in patients with severe BI [15]. Harbarth et al. measured PCT levels in patients with systemic inflammatory response syndrome (SIRS), sepsis, severe sepsis, and septic shock. PCT levels were found to be significantly different among the four groups, and more severe infections were found to be associated with higher PCT values [16]. Simon et al. demonstrated that PCT was more effective than CRP for differentiating BIs in patients with SIRS admitted to the pediatric intensive care unit [17]. Moreover, Casado et al. concluded that PCT was superior to CRP or WBC count in children with sepsis, even in infants [18]. A study conducted on infants observed better sensitivity and specificity for PCT than CRP [7]. In pediatric patients with suspected UTIs, PCT was effective in differentiating upper UTIs [8]. In a previous study that compared VI and BI in children (ages: 1–36 months) who visited the emergency room with fever, CRP and PCT exhibited similar sensitivity for the diagnosis of BI; however, the specificity of PCT was better. In differentiating IBI, PCT showed better sensitivity and specificity than CRP [1].

Conversely, some studies have argued that there was no significant difference in the efficacy of PCT compared with those of other inflammatory indicators, especially CRP. According to a study wherein non-infectious patients and infectious patients admitted to ICU were compared, PCT had lower sensitivity, specificity, and AUC than CRP, and the authors concluded that PCT had low-diagnostic efficacy [19]. In addition, some studies revealed that among children who visited the emergency room with fever, CRP was more convenient and sensitive as an indicator for predicting severe BI in comparison with PCT [20]. A study on malnourished children also concluded that CRP was a useful marker to identify patients at risk for death [21]. Additionally, in another study, PCT was more sensitive and specific than CRP in patients with bacterial sepsis [12]. A study, conducted in febrile children under the age of 3 years showed that PCT was a more accurate marker than WBC, absolute neutrophil, and band count in children with serious BI [22].

The limitations of this study are that it was conducted in a single center. As such, the number of enrolled patients was limited. Secondly, the number of patients per group and the diagnoses were uneven owing to the retrospective nature of the study. Lastly, there may be a possibility of misclassification of bacterial respiratory infections as VI, especially single boca-viral infections. However, all four patients who were diagnosed with pneumonia or bronchiolitis improved without the administration of antimicrobials. However, this study has strengths in that BIs were divided into four groups, and analyses were conducted between their classified subgroups. In addition, we compared and analyzed the findings from patients with inflammatory diseases in a manner similar to previous studies.

This study utilized retrospective analysis. Accordingly, there was a possibility of selection bias as the PCT tests were not performed in all patients. Additional, well-designed studies are required to identify newer and more accurate biomarkers to discriminate between BI and IBI prospectively.

In conclusion, a comparison of mean values revealed that PCT and ESR were the most useful inflammatory markers to differentiate BI from NBI and IBI from other types of infection. Though the PCT value proved to be useful, it had poor accuracy.

## Figures and Tables

**Figure 1 children-09-01263-f001:**
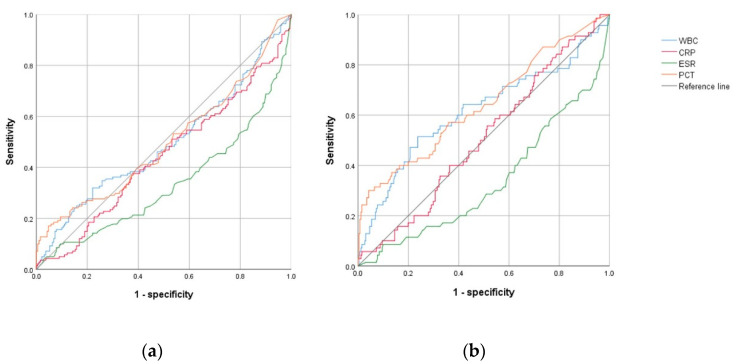
Comparison of receiver operating characteristic curve analyses of inflammatory markers between the (**a**) bacterial infection and nonbacterial infection group, and between the (**b**) invasive bacterial infection group and all other groups. Abbreviations: WBC, white blood cell; CRP, C-reactive protein; ESR, erythrocyte sedimentation rate; PCT, procalcitonin.

**Table 1 children-09-01263-t001:** Comparison of demographics between bacterial infection and non-bacterial infection groups.

	BI, n = 205 (%)	NBI		*p*
		VI, n = 118 (%)	ID, n = 151 (%)	
Sex, female	83 (40.5)	114 (42.4)		0.679
		49 (41.5)	65 (43.0)	0.889
Age, years †	4.2 ± 5.0	3.4 ± 3.0		0.057
		2.9 ± 2.6	3.8 ± 3.2	0.027
Hospitalization days †	8.7 ± 10.1	5.3 ± 2.7		<0.001
		5.1 ± 3.1	5.5 ± 2.4	<0.001

Abbreviations: BI; bacterial infection, NBI; nonbacterial infection, VI; viral infection, ID; inflammatory disease. † Theses variables are presented as means ± standard deviations.

**Table 2 children-09-01263-t002:** Comparisons of inflammatory markers among studied groups.

	WBC 10^6^/L	CRP mg/L	ESR mm/h	PCT ng/mL
Bacterial infection (BI)	13,177 ± 6484	61.4 ± 69.8	**39.1 ± 32.4**	**4.2 ± 16.9**
IBI	**14,797 ± 7148**	69.3 ± 77.6	**35.3 ± 30.3**	**8.1 ±** **23.8**
MBI	11,976 ± 5127	36.8 ± 44.8	44.5 ± 30.9	0.4 ± 0.8
TBI	11,241 ± 4350	41.9 ± 69.3	34.0 ± 32.7	2.6 ± 10.8
LBI	11,941 ± 6203	69.0 ± 15.1	45.3 ± 36.7	**0.6 ± 1.1**
Nonbacterial infection (NBI)	**12,959 ± 5841**	62.2 ± 60.1	**54.8 ± 28.2**	**1.1 ± 2.5**
VI	11,767 ± 6207	40.7 ± 51.5	38.3 ± 21.4	1.0 ± 3.3
ID	13,890 ± 5378	79.0 ± 61.1	61.4 ± 28.0	1.1 ± 1.7

Abbreviations: WBC; white blood cell, CRP; C-reactive protein, ESR; erythrocyte sedimentation rate, PCT; procalcitonin, IBI; invasive bacterial infection, MBI; mucosal bacterial infection, TBI; toxigenic bacterial infection, LBI; localized bacterial infection, VI; viral infection, ID; inflammatory diseases. Bold numbers indicated the statistical differences.

**Table 3 children-09-01263-t003:** Comparison of area under curve values between bacterial infection and nonbacterial infection groups.

	AUC	95% CI	*p*	Cut-Off	Sensitivity	Specificity
BI–NBI						
WBC	0.495	0.430-0.560	0.876	16530	71.2	22.7
CRP	0.443	0.380–0.506	0.075	4.1	20.8	89.5
ESR	0.335	0.274–0.395	<0.001	34.0	54.6	74.4
PCT	0.498	0.433–0.563	0.958	4.6	84.9	5.6
IBI–other groups						
WBC	0.615	0.532–0.698	0.003	15860	46.8	76.3
CRP	0.508	0.434–0.581	0.843	45.2	55.3	52.8
ESR	0.336	0.262–0.411	<0.001	44.0	68.6	58.4
PCT	0.640	0.562–0.719	<0.001	1.8	39.4	87.1

Abbreviations: AUC; area under the curve, CI: confidence interval, BI; bacterial infection, NBI; nonbacterial infection, WBC; white blood cell, CRP; C-reactive protein, ESR; erythrocyte sedimentation rate, PCT; procalcitonin.

## Data Availability

Not applicable.

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
