# Peer review of "Usefulness of Procalcitonin in the Diagnosis of Bacterial Infection in Immunocompetent Children"

_children, 2022, doi:10.3390/children9081263_

Round 1
Reviewer 1 Report
Dear author please try to put into a modern paradigm this research, I suggest a comparition with some new laboratory markers in inflammation or infection (Citok. or Interleukine...)
Author Response
- Dear author please try to put into a modern paradigm this research, I suggest a comparition with some new laboratory markers in inflammation or infection (Citok. or Interleukine...)
→ Thank you for your comment. This is a retrospective study so that we could not review the new inflammatory biomarkers. We tried to find the cut-off value and the usefulness as a biomarkers commonly prescribed in clinical practice. As the statistical power is too low, we will design a prospective study with other new biomarkers like your suggestion.
Reviewer 2 Report
The study by Park et al is a retrospective comparative analysis of inflammatory biomarkers levels in 491 children hospitalized due to presumed infections, categorized according to etiology (viral/bacterial) comparing procalcitonin (PCT) with CRP, ESR and WBC.
Given the retrospective nature of the study, there is a risk of introducing selection bias, but despite that, the data material has some value and could provide relevant information if analyzed, presented and discussed properly. However, as the manuscript currently stands, the results make little sense from a clinical perspective and it is not unexpected that the accuracy reported for all studied biomarkers is extremely low.
For the study to merit publication, the authors need to detail the selection process, thoroughly revise the categorization to make the groups less heterogeneous, better harmonize with clinical praxis and recognize the fact that a large proportion of children have mixed viral-bacterial infections or undetermined etiology.
Major comments:
-
The aim of the study is a bit vague. It is not clear if you aim to assess the ability of PCT to rule in/rule out severe bacterial infections at a specific cut-off or rather compare PCT levels between children with different infections.
-
The current classification is confusing and needs to be revised. This should be done either using established categories from the literature (E.g. Nijman et al Front Pediatr 2021) or by categorizing into more clinically relevant groups preferably also weighing in the uncertainty of the microbiological diagnosis (at least in a sensitivity analysis where only microbiologically confirmed infections are considered). For instance, the group of invasive BI mostly consists of children with urinary tract infection, which is not an invasive bacterial infection. In contrast the group of “Localized BI” also includes severe infections (intra-abdominal infections, mastoiditis).
-
The group of viral infection is not clearly described but seems to have been liberally defined as any detection of virus by PCR, despite the fact that detection of certain respiratory viruses by PCR is common in asymptomatic children (Rhedin et al Pediatrics 2014).
-
There are no children classified as having mixed viral-bacterial infection despite the increasing evidence that a large proportion of respiratory infections in children are mixed viral-bacterial infections (Bogaert et al
-
There is a significant difference in age between the groups that needs to be dealt with in the analyses.
-
Please describe the selection procedure in more detail. How many children were hospitalized for infections during the study period? Were any children excluded? This is important for evaluating potential selection bias.
Minor comments
ABSTRACT
-
Please present the key results in the abstracts, there are currently no numbers presented.
-
P1, R11: “is necessary for BIs” change to “is sometimes necessary for BIs”
-
P1, R21: spell out the abbreviation IBI the first time you introduce it
INTRODUCTION
-
P1, r28-30: Please soften this statement with “for many conditions” or similar as not all bacterial infections are hard to differentiate from viral infections.
-
P1, r36: Please change “Since 1993, PCT has been presented…” to “During the last decades, PCT has gradually been introduced in clinical praxis” or similar
METHODS
-
P2, r 68-76: “All patients underwent…” The sampling procedures are presented as if this was a prospective study, which is misleading. Please change to “All patients that were sampled with… were included.” and “Sampling was performed according to clinical practice…, where the majority of children are sampled with…” or similar.
-
P2, R79-80: “Cases with a WBC count…” How were urinary samples collected in infants? Bladder puncture? This is important as bag tests etc are easily contaminated in infants.
RESULTS
-
P3, R 11-112: The group of mucosal BI appears to mainly consist of children with upper respiratory tract infections that usually don’t need hospitalization? Were there any complicating factors here?
-
P5, R182-186: “...were determined to be useful markers…” This statement is not supported by your data as the AUCs were extremely low.
DISCUSSION
-
The whole manuscript, but in particular the discussion, would be improved by general language revision.
-
Table 2 - Consider presenting this data in a figure to facilitate the interpretation.
Author Response
Major comments:
- The aim of the study is a bit vague. It is not clear if you aim to assess the ability of PCT to rule in/rule out severe bacterial infections at a specific cut-off or rather compare PCT levels between children with different infections.
→ As mentioned, this study aims to evaluate the usefulness of PCT in the diagnosis of BIs, especially IBI, in children by comparing it with other inflammatory markers (line 45-46, page 2). In the beginning of the study, we wanted to confirm the usefulness of PCT to diagnose with bacterial infection and suggest the cut-off value of PCT. However, the statistical power of PCT were low so that we could not suggest the cut-off values. The purpose of this study was confusing because the conclusion is that PCT in useful. Therefore, we modified the conclusion according to the aim of this study.
- The current classification is confusing and needs to be revised. This should be done either using established categories from the literature (E.g. Nijman et al Front Pediatr 2021) or by categorizing into more clinically relevant groups preferably also weighing in the uncertainty of the microbiological diagnosis (at least in a sensitivity analysis where only microbiologically confirmed infections are considered). For instance, the group of invasive BI mostly consists of children with urinary tract infection, which is not an invasive bacterial infection. In contrast the group of “Localized BI” also includes severe infections (intra-abdominal infections, mastoiditis).
→ Thank you for comments. According to the good and useful reference who recommended, we rechecked the definitions of grouping. Positive culture studies from blood, urine, and CSF were defined as the invasive bacterial infection in the reference (Appendix A. from Nijman et al Front Pediatr 2021). We did not focus on the serious bacterial infections like the reference (Nijman et al Front Pediatr 2021) but, focused on BI group, especially IBI. We checked the definitions of groups once again with the recommended reference, and adjusted the inclustion/exclusion criteria.
Appendix A. Overview of serious bacterial infections and the varying reference standards used in diagnostic accuracy studies (Nijman et al Front Pediatr 2021)
|
|
|
|
Type of sample |
Definition of Positive test result |
Example of clinical diagnosis |
|
Serious infections |
Serious bacterial infections (SBI) ^ |
Invasive bacterial infections |
Blood culture |
Single bacterial pathogen isolated |
Bacteraemia, sepsis |
|
CSF culture |
Single bacterial pathogen isolated (+- CSF pleocytosis) |
Bacterial meningitis |
|||
|
Urine culture |
Single bacterial pathogen isolated, monoculture; (+- urine microscopy and/or urinalysis) |
Urinary tract infection, pyelonephritis |
|||
|
Blood PCR |
Single bacterial pathogen PCR positive |
Bacteraemia, sepsis |
|||
|
CSF PCR |
Single bacterial pathogen PCR positive |
Bacterial meningitis |
|||
|
|
Skin swab culture |
Single bacterial pathogen isolated |
Bacterial soft tissue infection.eg: Staphylococcal/Streptococcal infection |
||
|
Throat swab culture |
Single bacterial pathogen isolated |
Group A Streptococcal infection, scarlet fever |
|||
|
Stools culture |
Single bacterial pathogen isolated |
Bacterial gastro-enteritis, e.g. Salmonella, Shigella |
|||
|
Sputum culture |
Single bacterial pathogen isolated |
Bacterial upper or lower respiratory tract infection |
|||
|
Broncho-alveolar lavage culture |
Single bacterial pathogen isolated |
Bacterial pneumonia |
|||
|
Wound swab culture |
Single bacterial pathogen isolated |
Bacterial soft tissue infection |
|||
|
Bacterial cultures other |
Single bacterial pathogen isolated; e.g. from drained fluid form chest, abscess, or joint |
Septic arthritis, bacterial empyema, abscess |
|||
|
Bacterial PCR other |
Single bacterial pathogen PCR positive |
Pertussis |
|||
|
Chest X ray |
Evidence of lobar pneumonia, i.e. focal consolidation, or effusion, or empyema |
Bacterial pneumonia |
|||
|
USS, CT, MRI, DMSA |
Evidence of serious bacterial infection |
abscess, preseptal / orbital cellulitis, mastoiditis, appendicitis, peritonitis, deep tissue infection, transverse myelitis/discitis, osteomyelitis; support diagnosis of UTI / pyelonephritis |
|||
|
Clinical diagnosis |
In the absence of any diagnostics |
Abscess, cellulitis, scarlet fever |
|||
|
Serology |
Positive for acute bacterial infection (IgG, IgM) |
Mycoplasma infection |
|||
|
Other |
Positive urine antigen |
Legionella; Pneumococcal infection (>5 years) |
|||
|
Other or atypical (non-viral) serious infection |
Histology |
|
Appendicitis |
||
|
Clinical criteria / consensus / echo |
|
Kawasaki |
|||
|
Chest XR / IGRA / Mantoux / other / consensus |
|
TB |
|||
|
CSF pleocytosis with predominant polymorphs |
No pathogen identified, with CSF obtained post-antibiotics |
Sterile or aseptic meningitis |
|||
|
Blood |
Serology or PCR |
Rickettsiosis |
|||
|
Fungus isolated from (non-) sterile site |
Culture or PCR |
Invasive or non-invasive fungal infection |
|||
|
Blood |
Parasitaemia and microscopy |
Malaria |
|||
|
Serious viral infections |
Any sample type |
PCR or culture |
Confirmed causative viral pathogen in serious illness in need for hospital admission and (intensive) supportive care |
||
- The group of viral infection is not clearly described but seems to have been liberally defined as any detection of virus by PCR, despite the fact that detection of certain respiratory viruses by PCR is common in asymptomatic children (Rhedin et al Pediatrics 2014).
→ Thank you for your comment. We also knew about the asymptomatic infection but, we did not perform the laboratory test in asymptomatic children in this study. We performed the diagnostic tests according to the presented symptoms and physical examinations. We did not perform the diagnostic laboratory tests not related symptoms. We described about the clinical protocol in Methods section (2.2. Study design).
- There are no children classified as having mixed viral-bacterial infection despite the increasing evidence that a large proportion of respiratory infections in children are mixed viral-bacterial infections (Bogaert et al).
→ Thank you for your comments. BI and VI groups were classified according to the causative microorganism regardless of the final diagnosis. As like your mentions, there were a few cases combined viral infection in mucosal bacterial infection and inflammatory diseases groups. Therefore, combined cases were excluded from this study because they were likely to affect the statistical results. We also described about these in Method section.
“Cases combined bacterial and viral etiologies were excluded. ID group included cases of autoimmune or non-infectious IDs. Cases in ID group were also excluded with positive bacterial or viral studies.”
- There is a significant difference in age between the groups that needs to be dealt with in the analyses.
→ Thank you for your comment. We analyzed and added the description according to the ages in Results section.
There were significant differences according to the age groups between BI and NBI groups and among BI subgroups (all p<0.001). In infants, only ESR was different between BI and NBI (31.4±27.6 vs 48.1±24.6 mm/h, p=0.008) and between IBI and other groups (31.6±27.5 vs 46.1±25.7 mm/h, p=0.019). In toddlers, all biomarkers were not shown significant differences between BI and NBI as well as between IBI and other groups (all p>0.05). In preschoolers, ESR was different between BI and NBI (38.3±34.8 vs 58.1±29.5 mm/h, p=0.013) and WBC was different between IBI and other groups (19,295±4,280 vs 12,782±5,206 *106/L, p=0.015). In middle childhood and teenagers, there was no difference between BI and NBI as well as IBI and other groups (all p>0.05).
- Please describe the selection procedure in more detail. How many children were hospitalized for infections during the study period? Were any children excluded? This is important for evaluating potential selection bias.
→ Thank you for your comment. Those were very important information to decide whether selection bias or not. More than two thousand children were annually admitted in our hospital at Department of Pediatrics. We excluded the preterm babies who admitted in neonatal intensive care unit. Some patients did not want to perform PCR study because of the costs not covered by the insurance system so that these subjects were also excluded in this study. There were many mixed infections and many cases unclearly clarified the etiologies of infection through the culture or PCR studies. We only included the patients who were performed all biomarkers; WBC, CRP, ESR and PCT at the 1st day of admission. No repeated studies were included. They were excluded as well so that finally analyzed only 475 children despite of many admitted patients during the 7-years’ study period.
Minor comments:
ABSTRACT
- Please present the key results in the abstracts, there are currently no numbers presented.
→ Thank you for your comment. We edited the description of numbering and so on in Abstract.
- P1, R11: “is necessary for BIs” change to “is sometimes necessary for BIs”
→ Thank you for your comment. We edited the description of Abstract so that this phrase was deleted.
- P1, R21: spell out the abbreviation IBI the first time you introduce it
→ We already described BI fully as bacterial infection in line 12, page 1 as well as IBI fully as invasive BI in Abstract.
INTRODUCTION
- P1, r28-30: Please soften this statement with “for many conditions” or similar as not all bacterial infections are hard to differentiate from viral infections.
→ Thank you for your comment. We added “for many conditions” and corrected the mentions smoothly as you suggested in Introduction section.
- P1, r36: Please change “Since 1993, PCT has been presented…” to “During the last decades, PCT has gradually been introduced in clinical praxis” or similar
→ Thank you for your comment. We edited the sentence to “During the last decades, procalcitonin (PCT) has gradually been introduced in clinical praxis as an inflammatory marker to differentiate BIs” in Introduction section.
METHODS
- P2, r 68-76: “All patients underwent…” The sampling procedures are presented as if this was a prospective study, which is misleading. Please change to “All patients that were sampled with… were included.” and “Sampling was performed according to clinical practice…, where the majority of children are sampled with…” or similar.
→ Thank you for your suggestion. We edited the description to “All patients that were sampled with PCT, CRP, WBC count, and ESR were included”.
- P2, R79-80: “Cases with a WBC count…” How were urinary samples collected in infants? Bladder puncture? This is important as bag tests etc are easily contaminated in infants.
→ Thank you for your comment. The sampling method for urine is very important. We described the sampling method “urine collected through aseptic catheterization or cystocentesis” in Method section.
RESULTS
- P3, R 11-112: The group of mucosal BI appears to mainly consist of children with upper respiratory tract infections that usually don’t need hospitalization? Were there any complicating factors here?
→ In this study, we enrolled the inpatients as well as outpatients visiting the emergency department or outpatients clinics. This inclusion criteria were described in Methods. As you said, the number of MBIs was small because MBIs were not frequently hospitalized and not performed frequently laboratory blood tests. However, there was no complicated cases through the medical chart reviews. Therefore, the mean hospitalized days was short in MBI group than other subgroups. Thank you for your comments.
This retrospective analysis was conducted on immunocompetent patients under the age of 18 years, who underwent a PCT test in the outpatient department, emergency department, or as inpatients on their day of admission in the Chung-Ang University Hospital between July 2012 and June 2019.
- P5, R182-186: “...were determined to be useful markers…” This statement is not supported by your data as the AUCs were extremely low.
→ Thank you for your comment. As you mentioned, all authors reinterpreted the results with low statistical powers and discussed about the conclusion, so that we revised the flow of discussion section and conclusions.
“In conclusion, PCT and ESR was the most useful inflammatory markers to differenti-ate BI from NBI and IBI from all others when put the comparison of the mean values. However, only PCT of them was shown useful with a poor accuracy. We need a further study to find more accurate biomarkers discriminating BI and IBI.”
DISCUSSION
- The whole manuscript, but in particular the discussion, would be improved by general language revision.
→ Thank you for your comment. We edited the manuscript by English professional editor again.
- Table 2 - Consider presenting this data in a figure to facilitate the interpretation.
→ Thank you for your comment. Instead of changing the table 2, we modified the table 2 to make it easier to interpret.
Reviewer 3 Report
This is a well-organized study about the role of serum procalcitonin (PCT) in the differential diagnosis among pediatric patients with a bacterial infection. However, the statistical power of PCT was insufficient to reveal new scientific applications, and there are also some issues that the authors need to address.
2. Material and Methods
Page 2, line 46: The population of this study was pediatric and adolescent patients under 18 years old. It is expected that a significant number of pediatric patients who come to university hospitals will be immunocompromised patients (eg. hematologic malignancies, HSCR, SOT, SCIDs patients.). Immunocompromised patients may show relatively severe symptoms even with the same bacterial infection, so the population distribution must be considered to conduct an exact analysis.
Page 2, line 49: The classification expressed as BI / IBI / MBI / TBI / LBI was arbitrarily classified by the authors? If the same classification method has been used in other literature, please provide a reference.
Page 2, line 61: Were there any BI or VI co-infection cases? For example, if there is a case of CMV enteritis with intra-abdominal infection, or cholangitis, which group should it be classified as BI or VI?
3. Results
Page 5, Table 2:
Match the punctuation marks with other sentences (IBI ─ other BI groups (please choose en dash or hyphen). And since there are too many p-values in the table, it is difficult to recognize the significant value at a glance. Please edit only the meaningful parts in bold font or mark them with an asterisk.
Page 6, table 3:
In my personal experience, SPSS statistical program cannot draw ROC for the combined two markers. And is it not possible to obtain sensitivity, specificity, and cut-off with this method? If a cut-off is not available, how can the combined two markers be used specifically in clinical practice?
Page 7, Figure 2: This figure should have been numbered as figure 3.
Author Response
This is a well-organized study about the role of serum procalcitonin (PCT) in the differential diagnosis among pediatric patients with a bacterial infection. However, the statistical power of PCT was insufficient to reveal new scientific applications, and there are also some issues that the authors need to address.
→ Thank you for your comment. All authors reinterpreted the results with low statistical powers and discussed about the conclusion, so that we revised the flow of discussion section and conclusions.
“In conclusion, PCT and ESR was the most useful inflammatory markers to differenti-ate BI from NBI and IBI from all others when put the comparison of the mean values. However, only PCT of them was shown useful with a poor accuracy. We need a further study to find more accurate biomarkers discriminating BI and IBI.”
Material and Methods
- Page 2, line 46: The population of this study was pediatric and adolescent patients under 18 years old. It is expected that a significant number of pediatric patients who come to university hospitals will be immunocompromised patients (eg. hematologic malignancies, HSCR, SOT, SCIDs patients.). Immunocompromised patients may show relatively severe symptoms even with the same bacterial infection, so the population distribution must be considered to conduct an exact analysis.
→ Thank you for your comment. In our hospital, there was no pediatric hemato-oncologist. The immune-deficient patients were transferred and followed up in a few medical centers in Korea. Thus, all participants in this study were immunocompetent. We added a detailed description of the immune status of the enrolled subjects.
(Title) Usefulness of procalcitonin in diagnosis of bacterial infection in immunocompetent children
(Methods) This retrospective analysis was conducted on immunocompetent patients under the age of 18 years
- Page 2, line 49: The classification expressed as BI / IBI / MBI / TBI / LBI was arbitrarily classified by the authors? If the same classification method has been used in other literature, please provide a reference.
→ Authors defined BI and NBI, especially IBI according to the reference and added the reference number [8]. However, MBI, TBI, and LBI were arbitrarily classified by authors according to the pathogenesis and characteristics of infected tissues.
- Page 2, line 61: Were there any BI or VI co-infection cases? For example, if there is a case of CMV enteritis with intra-abdominal infection, or cholangitis, which group should it be classified as BI or VI?
→ Thank you for your comments. BI and VI groups were classified according to the causative microorganism regardless of the final diagnosis. As like your mentions, there were a few cases combined viral infection in mucosal bacterial infection and inflammatory diseases groups. Therefore, combined cases were excluded from this study because they were likely to affect the statistical results. We also described about these in Method section.
“Cases combined bacterial and viral etiologies were excluded. ID group included cases of autoimmune or non-infectious IDs. Cases in ID group were also excluded with positive bacterial or viral studies.”
Results
- Page 5, Table 2: Match the punctuation marks with other sentences (IBI ─ other BI groups (please choose en dash or hyphen). And since there are too many p-values in the table, it is difficult to recognize the significant value at a glance. Please edit only the meaningful parts in bold font or mark them with an asterisk.
→ Thank you for your comment. We edited the table 2 with bold font in significant differences in Table 2.
- Page 6, table 3: In my personal experience, SPSS statistical program cannot draw ROC for the combined two markers. And is it not possible to obtain sensitivity, specificity, and cut-off with this method? If a cut-off is not available, how can the combined two markers be used specifically in clinical practice?
→ Thank you for your comment. Your experience with SPSS about the cut-off value was right. SPSS shows the sensitivities and specificities for each value, and does not provide an appropriate cut-off value. Therefore, the authors found the optimal cutoff values using Medcalc 12 and presented the sensitivities and specificities according to the optimal cut-off values by Youden index.
- Page 7, Figure 2: This figure should have been numbered as figure 3.
→ Thank you for your comment. We edited the numbering of tables and figures again.
Round 2
Reviewer 2 Report
Thank you for the opportunity to review this revised version of your manuscript. I appreciate that you have tried to align the categories with the previous literature by Nijman et al and the manuscript has significantly been improved. Nevertheless there are still some issues that need to be dealt with before the manuscript is suited for publication. Mostly, more details on the cohort need to be provided. (E.g. number of screened children, excluded children and specific diagnoses in the viral group). There are also some grammatical errors and the manuscript would benefit from professional language editing.
- Although the authors have provided more details on the study design, I still miss key numbers describing the cohort. Please state:
o The number of patients that fulfilled the inclusion criteria (“immunocompetent patients under the age of 18 years, who underwent a PCT test in the outpatient department, emergency department, or as inpatients on their day of admission in the Chung-Ang University Hospital between July 2012 and June 2019“)
o The number of patients that were excluded (e.g. mixed viral-bacterial infections, no PCR test performed etc). This is really important as your study cohort represents a highly selected group of patients.
- Please add details on how many patients that were hospitalized. You currently only mention the length of stay, which easily gives the impression that all children were hospitalized.
- Thank you for the response. Although you did not study asymptomatic children, the fact that some viruses are frequently detected in asymptomatic individuals makes it important to know how you defined viral infection. Which viruses are included in your respiratory PCR? Would a children with respiratory symptoms testing positive only for e.g. bocavirus or coronavirus HKU1 be classified as having viral disease regardless of CRP (even if really elevated?)? If so, you need to discuss the possibility of misclassification of bacterial respiratory infections as VI in the discussion section.
- P3, R 116-129: “The NBI group was subclassified into the VI (n=118; 43.9%) and ID groups (n=151; 56.1%). In the 117 IBI group, 4.3% (n=4) of patients had meningitis…“
There is inconsistency in how the numbers are presented. I prefer the way you present it for the IBI group (where the number and procentage is presented next to the diagnosis rather than as you do it for LBI with several numbers/percentage following each other. I also suggest that you present the diagnoses in the order starting with the most frequent diagnose rather than least frequent (as you currently do for the IBI group). Finally, please also provide details on diagnoses in the VI group.
- P6, R221-229: “PCT is a propeptide of calcitonin which is normally secreted from the C-cells of the…”
This first paragraph is better suited in the introduction as it provides background to the study rather than discussing the findings. To facilitate for the reader I suggest that the first sentence in the discussion summarizes the study/ key results. E.g. “In this retrospective study of children we compared blood biomarker levels in children with BI and NBI and report that…”
- P7, R 263-265: “Lastly, as this study was analyzed retrospectively, there is a possibility of selection bias as the PCT tests were not performed in all patients, but only in patients with more severe symptoms.”
Rather than just mentioning this as a limitation I would suggest that you add a speculative sentence on the potential impact of such bias. Wouldn’t a selected group of severe “text book” cases rather overestimate the performance of biomarkers rather than underestimate it? Please also socioeconomic status here in line with your point-by-point reply (some children lack health insurance covering the costs for PCT testing).
Author Response
Thank you for the opportunity to review this revised version of your manuscript. I appreciate that you have tried to align the categories with the previous literature by Nijman et al and the manuscript has significantly been improved. Nevertheless there are still some issues that need to be dealt with before the manuscript is suited for publication. Mostly, more details on the cohort need to be provided. (E.g. number of screened children, excluded children and specific diagnoses in the viral group). There are also some grammatical errors and the manuscript would benefit from professional language editing.
→ We edited again from professional language editing. Thank you for the comment.
- Although the authors have provided more details on the study design, I still miss key numbers describing the cohort. Please state:
- The number of patients that fulfilled the inclusion criteria (“immunocompetent patients under the age of 18 years, who underwent a PCT test in the outpatient department, emergency department, or as inpatients on their day of admission in the Chung-Ang University Hospital between July 2012 and June 2019“)
- The number of patients that were excluded (e.g. mixed viral-bacterial infections, no PCR test performed etc). This is really important as your study cohort represents a highly selected group of patients.
→ Thank you for the comment. We added the description about the number of patients according to the exclusion criteria in the 3.1. Results section.
“During the 7-year study period, 24691 immunocompetent children under 18 years of age visited our medical center. Among them, 3203 children underwent PCT tests in our emergency department (n = 350), outpatient clinic (n = 577), and hospitalized ward (n = 2,276). Overall, 1567 cases were revealed to be follow-up cases and were excluded. Ad-ditionally, 193 neonatal intensive care unit cases were excluded. Moreover, 952 children who did not undergo PCR tests were excluded due to unclarified etiologies. Furthermore, 17 cases were excluded because they were combined BI/VI cases or were IDs with VIs.”
- Please add details on how many patients that were hospitalized. You currently only mention the length of stay, which easily gives the impression that all children were hospitalized.
→ We mentioned about the number of admitted patients. “Among the 462 admitted patients, the mean duration of hospitalization was 6.7 ± 7.1 days.”
- Thank you for the response. Although you did not study asymptomatic children, the fact that some viruses are frequently detected in asymptomatic individuals makes it important to know how you defined viral infection. Which viruses are included in your respiratory PCR? Would a children with respiratory symptoms testing positive only for e.g. bocavirus or coronavirus HKU1 be classified as having viral disease regardless of CRP (even if really elevated?)? If so, you need to discuss the possibility of misclassification of bacterial respiratory infections as VI in the discussion section.
→ Thank you for the comment. We added the description about respiratory PCR panel using our hospital in 2.2. Method section. “for the detection of respiratory viruses (adenovirus, respiratory syncytial virus A/B, In-fluenza A/B virus, parainfluenza virus 1/2/3/4, rhinovirus A/B/C, metapneumovirus, en-terovirus, coronavirus 229E/NL63/OC43, and bocavirus 1/2/3/4) via PCT test.” We reviewed eCRF again and we found 3 coronaviral infections but, no single viral infection. However, 4 cases were revealed single bocaviral infection and their final diagnosis were bronchiolitis or pneumonia. Of 4 cases, three were found elevated CRP but, normal PCT. According to your suggestion, we added the description about the possibility of misclassification of bacterial respiratory infections as VI in the discussion section.
“Lastly, there may be a possibility of misclassification of bacterial respiratory infections as VI, especially single bocaviral infections. However, all four patients who were diagnosed with pneumonia or bronchiolitis improved without the administration of antimicrobials.”
- P3, R 116-129: “The NBI group was subclassified into the VI (n=118; 43.9%) and ID groups (n=151; 56.1%). In the 117 IBI group, 4.3% (n=4) of patients had meningitis…“
There is inconsistency in how the numbers are presented. I prefer the way you present it for the IBI group (where the number and procentage is presented next to the diagnosis rather than as you do it for LBI with several numbers/percentage following each other. I also suggest that you present the diagnoses in the order starting with the most frequent diagnose rather than least frequent (as you currently do for the IBI group). Finally, please also provide details on diagnoses in the VI group.
→ Thank you for the comment. As you suggested, the description was unified and more common diagnosis were described at first.
- P6, R221-229: “PCT is a propeptide of calcitonin which is normally secreted from the C-cells of the…”
This first paragraph is better suited in the introduction as it provides background to the study rather than discussing the findings. To facilitate for the reader I suggest that the first sentence in the discussion summarizes the study/ key results. E.g. “In this retrospective study of children we compared blood biomarker levels in children with BI and NBI and report that…”
→ Thank you for the comment. As you suggested, we corrected the order of paragraphs. We changed the summary of this study to the first paragraph of discussion section and the first paragraph in the discussion were changed to the introduction section.
- P7, R 263-265: “Lastly, as this study was analyzed retrospectively, there is a possibility of selection bias as the PCT tests were not performed in all patients, but only in patients with more severe symptoms.”
Rather than just mentioning this as a limitation I would suggest that you add a speculative sentence on the potential impact of such bias. Wouldn’t a selected group of severe “text book” cases rather overestimate the performance of biomarkers rather than underestimate it? Please also socioeconomic status here in line with your point-by-point reply (some children lack health insurance covering the costs for PCT testing).
→ Thank you for the comment. In Korea, all citizens are covered by the national health insurance system if they are not illegal residents. However, some test are non-benefits, including respiratory virus PCR panel. Thus, some patients refused the non-benefit test because of the costs. Also, we erased the last limitation and described as a reason for the need for prospective further study with new biomarkers.
“This study utilized retrospective analysis. Accordingly, there was a possibility of selection bias as the PCT tests were not performed in all patients. Additional, well-designed studies are required to identify newer and more accurate biomarkers to discriminate between BI and IBI prospectively.”

Reviewer 3 Report
Thank you for correcting the text in such a short period of time.
Reading the comments in response to the reviewer's question, this study deserves to be considered as a study that further expanded the clinical usefulness of procalcitonin.